# Lactic Acid and an Acidic Tumor Microenvironment suppress Anticancer Immunity

**DOI:** 10.3390/ijms21218363

**Published:** 2020-11-07

**Authors:** Joy X. Wang, Stephen Y.C. Choi, Xiaojia Niu, Ning Kang, Hui Xue, James Killam, Yuzhuo Wang

**Affiliations:** 1Vancouver Prostate Centre, Vancouver, BC V6H 3Z6, Canada; joywang@alumni.ubc.ca (J.X.W.); schoi@bccrc.ca (S.Y.C.C.); gjk@killamcordell.com (J.K.); 2Faculty of Dentistry, University of British Columbia, Vancouver, BC V6T 1Z3, Canada; 3Department of Urologic Sciences, Faculty of Medicine, University of British Columbia, Vancouver, BC V5Z 1M9, Canada; xniu@bccrc.ca; 4Department of Experimental Therapeutics, BC Cancer Agency, Vancouver, BC V5Z 1L3, Canada; cokang@bccrc.ca (N.K.); hxue@bccrc.ca (H.X.)

**Keywords:** lactic acid, cancer-induced immunosuppression, epigenetic reprograming, de-differentiation, anticancer immunotherapy, acidic tumor microenvironment

## Abstract

Immune evasion and altered metabolism, where glucose utilization is diverted to increased lactic acid production, are two fundamental hallmarks of cancer. Although lactic acid has long been considered a waste product of this alteration, it is now well accepted that increased lactic acid production and the resultant acidification of the tumor microenvironment (TME) promotes multiple critical oncogenic processes including angiogenesis, tissue invasion/metastasis, and drug resistance. We and others have hypothesized that excess lactic acid in the TME is responsible for suppressing anticancer immunity. Recent studies support this hypothesis and provide mechanistic evidence explaining how lactic acid and the acidic TME impede immune cell functions. In this review, we consider lactic acid’s role as a critical immunoregulatory molecule involved in suppressing immune effector cell proliferation and inducing immune cell de-differentiation. This results in the inhibition of antitumor immune responses and the activation of potent, negative regulators of innate and adaptive immune cells. We also consider the role of an acidic TME in suppressing anticancer immunity. Finally, we provide insights to help translate this new knowledge into impactful anticancer immune therapies.

## 1. Introduction

In the 1920s, Otto Warburg first described a phenomenon where cancer cells displayed an altered metabolism, obtaining energy through glycolysis at disproportionately high rates even under aerobic conditions [1]. Through aerobic glycolysis, glucose is converted into lactic acid instead of carbon dioxide as a by-product. In order to compensate for the inefficiency in adenosine triphosphate (ATP) production, cancer cells increase their glucose metabolism by regulating key transporters and enzymes [2]. Energy can also be derived through the glutaminolytic pathway, where abundant glutamine is metabolized instead of glucose. From both pathways, high amounts of lactic acid are produced and subsequently discharged into the extracellular space between cancer cells (i.e., the tumor microenvironment (TME)). This excessive and constant generation of lactic acid results in an acidic TME [1]. Lactic acid concentrations in the TME can be as high as 10–30 mM, whereas its concentration under normal physiological conditions is about 1.5–3.0 mM [3]. The pH can be as low as 6.0–6.5.

For many years, lactic acid was simply regarded as a waste product of cancer metabolism. However, it is now recognized that lactic acid and the resulting acidic TME have widespread effects on cancer biology, stimulating angiogenesis, cancer cell local invasion, and distant metastasis. In 2013, we proposed that the overproduction of lactic acid and the resulting acidity in the TME is also a key mechanism of immune escape by disarming immune cells in the TME [4]. Numerous recent studies have provided evidence supporting the functional role of lactic acid in inhibiting anticancer immunity [5,6,7,8], and various mechanisms explaining how lactic acid and the acidic TME impede immune cell functions have also been uncovered [6,9,10,11].

This review will first summarize the associations of innate and adaptive immune cells to human cancers. We will then describe the major effects of lactic acid and an acidic TME on immune cell functions. The literature surveyed here is mainly from this past decade, with special attention paid to the most recent discovery that lactic acid epigenetically regulates histone modifications [10]. Finally, we provide insights for achieving better, more-holistic cancer immunotherapy through targeting lactic acid production and transport.

## 2. Lactic Acid and Innate Immune Cells in the TME

A wide variety of innate immune cells in the TME play crucial roles in human cancers. These include monocytes/macrophages, natural killer (NK) cells, neutrophils, and dendritic cells (Figure 1). Accumulating evidence suggests that the presence of innate immune cells in the TME functionally contributes to cancer cell survival and growth [12]. We briefly describe how some of the most important innate immune cells are affected by lactic acid and the acidic TME.

### 2.1. Macrophages

Macrophages are derived from monocytes and are important effector cells of innate immunity. They migrate via the circulatory system into almost every tissue of the body [13,14,15]. In response to their microenvironments, macrophages can adopt either a M1 macrophage (M1) or M2 phenotype. M1 macrophages are pro-inflammatory and function as phagocytic cells. They can directly target cancer cells by phagocytosis or through the release of cytokines (e.g., TNFα). They can also indirectly target cancer cells by recruiting (e.g., with chemokines such as CXCL10) and activating (e.g., with cytokines such as IFN-γ and IL-12) other immune effector cells [16].

M2 macrophages are widely regarded as having the opposite function and are considered critical for tumor development and progression [13,17]. They can release various cytokines and growth factors to promote immune suppression (e.g., IL-10, IL-13, TGFß, CCL9, etc.), cancer cell invasion/metastasis (e.g., EGF, MMP, etc.), and angiogenesis (e.g., VEGF, TNFα, etc.) [18].

There is evidence suggesting that both an acidic TME and increased lactic acid can significantly affect macrophages. For example, low pH in the TME can independently alter macrophage phenotype and functionality [5]. In particular, lactic acid secreted by cancer cells has a critical signaling function in the TME to induce M2 polarization [11]. Furthermore, when incubating M1 and M2 macrophages at pH 7.4 or 6.8, M2 macrophages showed higher viability and better fitness in the lower pH than their M1 counterparts [17]. Expressions of pro-inflammatory M1 markers (e.g., iNOS, MCP1, IL-6) were also lower at an acidic pH while expressions of M2 markers (e.g., MRC1, arginase 1 (Arg1), chitinase-3-like protein) were higher [17]. Lactic acid taken up by macrophages can intrinsically promote the upregulation of the M2-marker arginase 1 (Arg1) and the neovascularization factor VEGF [19,20]. When lactate levels decrease, Arg1 expression decreases. Similarly, when lactate levels increase, Arg1 expression also increases [19,20]. Thus, lactic acid initiates the expression of homeostatic genes that have been traditionally associated with M2 macrophages.

Reducing lactic acid levels will likely reduce the M2-polarization of macrophages. For example, in prostate cancer, El-Kenawi et al. discovered that the acidic TME contributes to the M2-polarization of macrophages both in vitro and in vivo [5]. They activated macrophages in vitro at either a physiological pH 7.4 or at an acidic pH below 6.8 and observed enhanced expressions of CD206 and a range of M2-related genes (e.g., Arg1, CD14, IL1b) under acidic conditions. Furthermore, macrophages cultured in extracellular acidosis increased their release of cytokines and chemokines involved in angiogenesis and tumor progression (e.g., VEGF, M-CSF, CD14) [5]. Importantly, when the acidic pH was buffered back to a physiological level in vivo by subcutaneous injections of sodium bicarbonate, CD206 and Arg1 expression in tumor associated macrophages (TAMs) were significantly reduced [5]. This highlights how TME acidity has a direct impact on macrophage phenotype, skewing differentiation to the pro-tumor M2 phenotype. This process can be reversed when the acidity is buffered back to a physiological pH of 7.4.

Given the consensus that TAMs are mainly cancer-promoting M2 macrophages, a number of anti-TAM drug candidates are in preclinical development and clinical trials [21].

### 2.2. Natural Killer Cells

Natural killer (NK) cells are innate immune cells that display rapid and potent cytolytic activity in response to transformed cells [22], and have a well-documented anti-tumor effect [23,24]. They participate in early tumor immune surveillance by producing and releasing perforin, granzymes, and cytokines. The therapeutic value of allogeneic NK cells was first observed in hematological cancers [25], and is being explored in solid tumors [26,27].

Lactic acid-induced extracellular acidosis in the TME inhibits the anti-tumoral activity of NK cells. Studies in melanoma mouse models show that decreasing the TME pH to 5.8–7.0 decreased the release of lytic granule contents, such as perforin and granzymes. It also decreased the secretion of IFN-γ and TNF-α, consequently decreasing the cytotoxic response against tumor cells [7]. In mouse NK cells, lactic acid reduces the expression of IFN-γ at both the mRNA and protein levels, and IFNγ production is completely inhibited at 15 mM lactic acid. This indicates that lactic acid alone can diminish cytokine production [7]. Similarly, increasing lactic acid levels in pancreatic cancer mouse models decreased NK cell activity and increased tumor size [8].

The effects of an acidic TME on NK cells are reversible across various cancers. By orally delivering bicarbonate in lymphoma mouse models and thus increasing the TME pH from an acidic 6.5–6.9 to a more physiological pH of 7.2–7.5, IFN-γ production by NK cells increased and tumor growth was delayed [28]. Moreover, Long et al. discovered that downregulation of the lactate transporter MCT4 in breast cancer cells promoted the cytotoxicity of host NK cells in vivo [29], which is consistent with our unpublished studies in prostate cancer (Choi and Niu). The downregulation of MCT4 resulted in decreased extracellular lactic acid concentrations and increased pH in the TME [29]. By buffering lactic acid and reversing the TME acidity, NK cells were observed to have enhanced activation and degranulation, as evidenced by more perforin and CD107a expressions [29]. Thus, NK cell effector functions are not only inhibited by lactic acid in the acidic TME, they can also be reacquired upon pH reversal.

### 2.3. Neutrophils

Neutrophils make up 50–70% of myeloid-derived white blood cells in human blood and are mainly involved in immunity against invading pathogens via cytokine secretion and phagocytosis [30]. While the role of neutrophils in cancer biology is still debated, recent studies suggest that these cells are actively involved in cancer progression and metastasis [31]. In 2009, Fridlender et al., first proposed that neutrophils exhibit “N1” and “N2” phenotypes like their macrophage counterparts [32]. The traditional, tumor-cytotoxic N1 phenotype has the potential to kill tumor cells, due to elevated levels of immune-activating factors (e.g., TNF-α, ICAM-1, FAS) [32] and direct antibody-dependent cytotoxicity [33,34]. On the other hand, the N2 phenotype is characterized as having higher expressions of arginase and pro-tumor factors (e.g., CCL2, CCL5, cathepsin G, neutrophil elastase) that induce immunosuppression in the TME [31]. Therefore, the functions of neutrophils within the TME seem to fall under distinct subsets, with N2 neutrophils being the pro-tumorigenic phenotype.

Evidence suggests that extracellular acidosis from cancer-generated lactic acid secretion acts as a key regulator of neutrophil apoptosis and function [35]. As observed by flow cytometry and Wright staining, neutrophil apoptosis is delayed when extracellular acidosis causes a decrease in intracellular pH [35,36]. The activation of caspase 3, a key regulator of apoptosis, is also reduced under such conditions [35]. This reduction in intracellular pH as caused by an acidic TME also alters the activity of various intracellular enzymes. Cao et al. discovered that an acidic environment promotes an alternative functional profile in neutrophils and is characterized by suppressed reactive oxygen species production and poor phagocytic ability [35]. Importantly, lactic acid in the acidic TME promotes the differentiation of neutrophils into its N2 phenotype [35,37]. Tumor associated neutrophils (TANs) in the acidic TME express high levels of β*2* integrin and CD11b/CD18, and they contribute to cancer growth and metastasis through multiple mechanisms including T-cell suppression, production of angiogenic factors, and secretion of proteases (e.g., MMP-9 and elastase) [37]. Although the effects of lactic acid on neutrophils and the precise mechanisms of action have yet to be fully characterized, it is evident that TME acidity significantly enhances the tumor-promoting functions of TANs.

The influential role of neutrophils in cancer biology and their potential as therapeutic targets are increasingly recognized. This may open up new opportunities for therapeutic interventions [38,39,40,41].

### 2.4. Dendritic Cells

Dendritic cells (DC) link innate immunity with the adaptive immune system and are responsible for activating adaptive immune responses. Monocytes have the capacity to differentiate into monocyte-derived DCs and can also function as macrophage precursors. They therefore play a critical role in the activation of naïve T cells and the induction of antigen-specific T cell-mediated immunity [42,43]. Since T cells are essential to an antitumor immune response, adequate DC functions are required for sufficient T-cell activation. However, the function of cancer-associated DCs is suppressed in the acidic TME [12].

The TME is rich in immunosuppressive factors that limit the immunostimulatory capacity of DCs [44]. Exposure to high levels of lactate (i.e., 40 mM) were found to inhibit DCs from differentiation and maturation [45]. When cultured with IL-4 and GM-CSF secreted from different tumor cell lines, DC precursors do not express CD1a and are unable to differentiate into DCs [45]. Thus, lactic acid-induced acidosis impairs monocyte differentiation into DCs. As with other innate immune cells, the inhibitory effects of lactic acid can be reversed when lactic acid production is blocked [45].

One approach to overcoming cancer immunosuppression for cancer immunotherapy is to enhance DC functions. In that regard, inhibition of IDO and STAT3 are being explored in mice and in clinical trials [42].

### 2.5. Myeloid Derived Myeloid Suppressor Cells

Under physiological conditions, bone marrow-derived myeloid suppressor cells (MDSCs) differentiate into granulocytes, macrophages, and DCs. This differentiation is impaired under acidic conditions, leading to an accumulation of MDSCs [46]. They are capable of inducing strong immunosuppressive effects through the expression of various cytokines and immunoregulatory molecules. MDSCs have been shown to inhibit lymphocyte homing, stimulate other immunosuppressive cells, deplete metabolites critical for T cell functions, express ectoenzymes that regulate adenosine metabolism, and produce reactive oxygen species [46]. These MDSCs accumulate in both experimental and clinical tumors and are considered strong contributors to the immunosuppressive TME. [44]. They remain a major obstacle for many cancer immunotherapies [44].

In the acidic TME, MDSC activity increases via the lactic acid-induced HIF1α pathway, resulting in increased programmed death-ligand 1 (PD-L1) expression and myeloid cell death [47]. Furthermore, MDSCs can initiate formation of the premetastatic niche by increasing angiogenesis and enhancing tumor cell stemness [48].

Accumulating evidence suggests that MDSCs can be a therapeutic target [49]. In fact, several ongoing clinical trials that aim to indirectly impact MDSC functions by targeting Arg1, iNOS, and STAT3 are ongoing in different cancer types [12].

## 3. Lactic Acid and T Cells in the TME

### 3.1. Cytotoxic T Cells

Cytotoxic T cells, or cytotoxic T lymphocytes (CTLs), are key players in antitumor immunity, because of their ability to selectively recognize and kill cancer cells [23]. T cells are fully activated when the T cell receptor recognizes a cancer peptide-MHC complex, and when additional co-stimulatory receptors are engaged [50]. The formation of this immunological synapse causes the release of perforin, granzymes, and immune-stimulating cytokines. Perforin forms pores on the target cancer cell membrane and creates an osmotic imbalance, allowing granzymes to enter the cell and cause protein degradation and apoptosis. Meanwhile, immune-stimulating cytokines (e.g., IL-2, IFNγ, and TNFα) help sustain the immunological response and develop long-term immunological memory [51].

The acidic TME created by cancer-generated lactic acid has multiple impairing effects on CTLs, including impaired chemotaxis and respiratory activities [52]. In tumor-bearing mice with a pathological TME pH of 6.0–6.5, a substantial reduction in CTL cytolytic activity and cytokine secretion was observed [53]. One potential mechanism is that lactic acid decreases CTL recruitment into the TME, while trapping T cells that are already present by impairing chemotaxis. The acidic TME further inhibits the functions of infiltrating CTLs already in the TME. Upon internalization of lactic acid through the SMCT2 transporter, CD4+ helper T cells and CTLs have either *phosphofructokinase* inhibited or Hexokinase 1 downregulated, resulting in the inhibition of glycolysis and the reduction of cell motility [54]. As such, T cells lose their responsiveness to chemokines and no longer migrate throughout the body.

The effector functions are reversible in both human and mouse CTLs when physiological pH is restored at the tumor site [53].

### 3.2. Regulatory T Cells

Regulatory T cells (Tregs) suppress abnormal and excessive immune responses to self- and non-self-antigens and help maintain immune homeostasis [55]. In cancer immunity, Tregs support tumor development and progression by inhibiting antitumor immunity [56,57]. Unlike the other immune cells described in this review, the activity and recruitment of Tregs are increased in the acidic TME. This further suppresses anticancer immunity and represents a major obstacle to effective anticancer immunotherapy [56,57]. The acidic TME also contributes to Treg induction. Indoleamine 2,3-dioxygenase is an immunoregulatory enzyme expressed by Tregs that convert tryptophan into kynurenine. Elevated levels in the acidic TME reduce tryptophan levels, which in turn activate stress response pathways that maintain Treg suppressive functions [58,59]. Overall, increased Treg activity in an acidic TME contributes to a reduction in the anticancer immune response.

## 4. Mechanisms Used by Lactic Acid to Suppress Anticancer Immunity

Lactic acid is no longer considered a waste product of the Warburg effect. Lactic acid in the TME is responsible for suppressing anticancer immunity. A number of recent studies provide evidence explaining how lactic acid impedes immune cell functions. In this section, we summarize its role in (1) inhibiting immune cell proliferation and survival [60]; (2) inducing immune cell de-differentiation [10]; and (3) the signaling of downstream processes [4,61] (Figure 2).

### 4.1. Lactic Acid Inhibits Immune Cell Proliferation and Survival in the TME

Immune cells typically exploit aerobic glycolysis to maintain their high proliferation rates during an active immune response. They thus utilize the Warburg effect to keep intracellular glycolytic intermediates at a high level to support cell proliferation [60]. Using T cells as an example, the conversion of glucose into lactic acid in the presence of oxygen is considered the characteristic metabolic switch, whereby naïve T cells proliferate and differentiate into effector T cells [60]. Proliferative T cells using glycolysis export lactate through the monocarboxylate transporter 1 (MCT1) [62]. However, because the export of lactic acid is concentration-gradient dependent, high levels of lactic acid in the TME block MCT1 activity, thus preventing the discharge of lactic acid and inhibiting T cell proliferation. Concentrations of lactate above 20 mM can reduce the number and activity of CTLs and NK cells by causing apoptosis both in vitro and in vivo [7]. Taken together, this explains both the reduction of CTL and NK cell numbers and their impaired functions in tumors with high concentrations of lactic acid. Interestingly, sodium lactate at a neutral pH does not exert such inhibitory effects on CTLs. Considering that MCTs cotransport lactate and protons according to the concentration gradient, both high lactate and low pH in the TME seem to be necessary to metabolically block CTLs [60].

### 4.2. Lactic Acid Induces De-Differentiation of Immune Cells in the TME

Recently, an exceptional study reported that lactic acid serves as an epigenetic regulator of gene transcription through histone modifications [10]. This is the first report of lactic acid directly affecting gene transcription at a global level. This discovery highlights not only how lactic acid can have a global impact in suppressing both innate and adaptive immunity, it also offers a possible mechanism by which other cancer-promoting pathways that increase angiogenesis, migration, and metastasis may be activated.

Lactic acid directly binds to histone lysine lactylation (Kla) sites, stimulating the expression of many genes in immune cells [10]. A total of 28 Kla sites have been identified experimentally using high-performance liquid chromatography (HPLC) tandem mass spectroscopy [10]. For example, the ability of M1 macrophages to polarize into the M2 phenotype in lactic acid-induced extracellular acidosis is likely due to histone lactylation at these Kla sites, thereby directly enhancing inflammation-independent biological pathways [10].

The discovery of these 28 Kla sites suggests that lactic acid can play a role in altering gene expression in additional cell types. For example, this may explain why multiple cell types are dedifferentiated in an acidic TME. It may also explain why so many immune cells lose their anticancer abilities in the acidic TME. Seeing lactic acid as a global epigenetic regulator allows for a more comprehensive understanding and a greater acceptance of its profound role in cancer biology, particularly in shaping anticancer immunity.

### 4.3. Lactic Acid Acts as a Signaling Molecule

Lactic acid is described as a key signaling molecule that plays a pivotal role in cancer cell migration, invasion, growth, angiogenesis, and immune escape [4,61]. This signaling function depends at least partially on lactic acid receptors. For example, G protein-coupled receptor GPR81 is one such lactic acid receptor found on both immune cells [63,64] and cancer cells [65]. Its activation in cancer cells promotes proliferation, drug resistance, and enhanced expression of PD-L1 [66,67,68]. In addition to this autocrine role, Brown et al. reported that lactic acid aids tumor growth in a paracrine fashion when GPR81 is activated on immune cells [69]. For example, activation of GPR81 on DCs is associated with decreased levels of cAMP, IL-6, and IL-12. This suggests that lactic acid signaling in DCs prevents the presentation of tumor-specific antigens to other immune cells [69]. Knockdown of GPR81 in mice suppressed the generation of Tregs [64].

## 5. Perspectives

Despite much research, success rates in treating advanced cancers have changed relatively little [70]. One reason for this is that most research is highly specific and thus narrowly focused, often neglecting more widely fundamental aspects of cancer biology. In the last decade, cancer immunotherapy has become one of the most promising types of treatment. However, currently available immunotherapies focus only on restoring or enhancing isolated components of the immune system, often being directed at a single immune cell type. For example, there are over 2000 clinical trials investigating PD-1/PD-L1-targeted drugs [71], yet only a minority of patients will respond to these agents [72].

Lactic acid is considered a key “oncometabolite” that plays an important role in cancer biology both directly and through its acidification of the TME. It can promote angiogenesis, cancer cell migration, and metastasis [73]. Furthermore, the excessive production of lactic acid by cancer cells and the resultant acidification of the TME suppress a whole host of innate and adaptive immune cells, leading to a critical hallmark of cancer—evasion of anticancer immunity. This is manifested in the inhibition of immune cell proliferation, induction of de-differentiation via epigenetic regulation, and alteration of cell functions via autocrine and paracrine signaling. By overcoming immune surveillance in low pH environments, cancer cells become better adapted for survival and metastasis.

We have previously reported that the role of lactic acid in cancer cell immune evasion seems to be evolutionarily conserved. It is also a mechanism utilized by pathogenic bacteria, endoparasites, and virus-infected host cells [74].

Lactic acid has complex effects on a wide spectrum of innate and adaptive immune cells that contribute to anticancer immunity. As described in this review, the effector functions that are inhibited by lactic acid and an acidic TME have been experimentally demonstrated as reversible in a variety of immune cell types across different cancers. Thus, if the acidic TME can be buffered back to a physiological condition, the anticancer functions of various immune cells can likely be restored. This has the potential to become an extremely powerful form of immunotherapy.

While such a generic hypothesis may sound simple to achieve, the actual specifics with respect to buffering the TME can be challenging. While some in vivo models show that ingestion of bicarbonate can restore the TME pH to a physiological level [28], whether such a strategy can be applied to patients remain unclear. A more hopeful strategy would be to therapeutically target major lactic acid generation pathways by inhibiting both glycolysis and glutaminolysis. However, given the fluid nature of cellular metabolism, the diversity of enzymes involved, and the abundance of alternative pathways, finding a sufficiently critical target remains elusive. Efforts specifically inhibiting glucose metabolism have yet to yield much success, and concerns over general toxicity to otherwise normal cells remain. Finally, inhibiting lactate transport from cancer cells into the TME by blocking lactic acid transporters could be another approach [75,76,77,78,79]. Ideally, if lactic acid can neither be produced by cancer cells nor transported into the TME, the acidity will gradually return to a physiological pH. Therapeutic manipulations directly targeting lactic acid production or transport could potentially be an extremely effective form of immunotherapy resulting in the restoration of anticancer immunity.

## Figures and Tables

**Figure 1 ijms-21-08363-f001:**
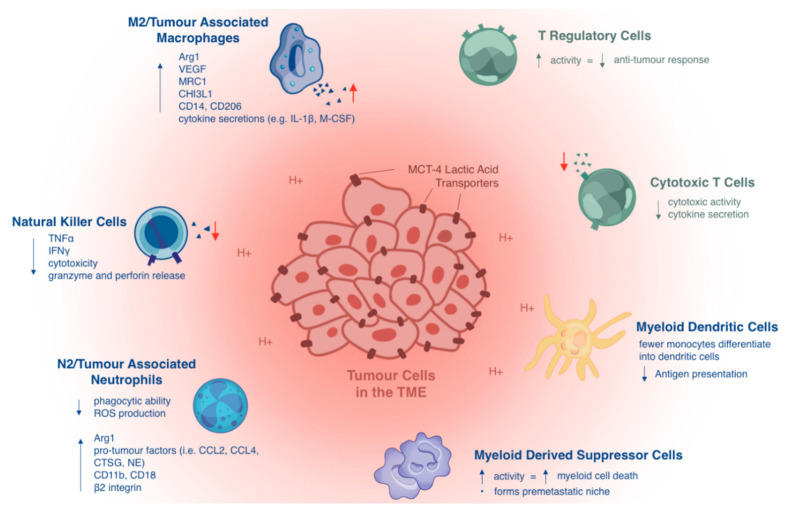
Lactic acid in the tumor microenvironment suppresses the antitumor immune response by negatively regulating innate and adaptive tumor-infiltrating immune cells. First, lactic acid impairs monocyte differentiation into dendritic cells (DCs) and further decreases their antigen-presentation functions. Second, lactic acid inhibits the antitumor activities of immune effector cells, including natural killer and cytotoxic T cells. Lastly, lactic acid promotes the infiltration of immunosuppressive cell types, such as M2 macrophages (M2) -like tumor-associated macrophages, N2 neutrophils (N2)-like tumor-associated neutrophils, myeloid-derived suppressor cells, and regulatory T cells, which can effectively inhibit the antitumor immune response and contribute to cancer immune escape.

**Figure 2 ijms-21-08363-f002:**
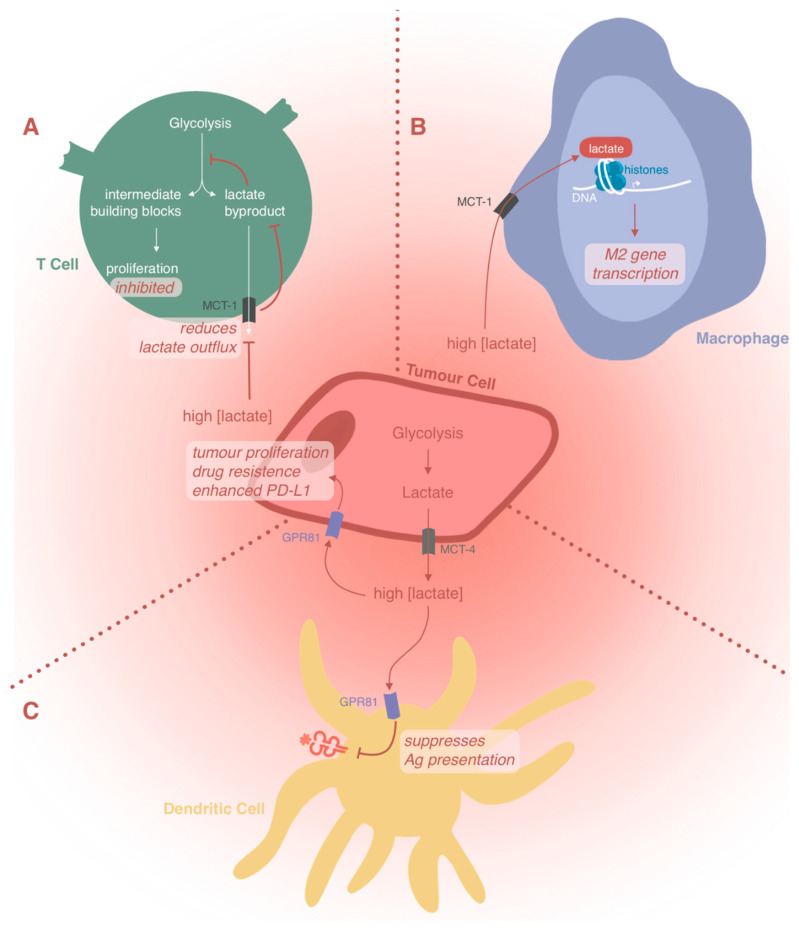
The molecular mechanisms by which lactic acid modulates immune cell responses. Lactic acid promotes cancer cell proliferation, drug resistance, and enhanced expression of programmed death-ligand 1 (PD-L1). In addition, lactic acid is responsible for suppressing antitumor immunity. (**A**) High lactic acid concentrations in the tumor microenvironment disrupts the [H+] gradient between T cells and their environment, reducing monocarboxylate transporter 1 (MCT1)-mediated lactic acid export from T cells. This inhibits effector T cell proliferation. (**B**) Lactic acid acts as an epigenetic regulator and induces M2 macrophage polarization through epigenetic reprogramming. It directly binds to histone lysine lactylation (Kla) sites (28 of these have been experimentally identified) to direct downstream gene transcription, thereby inducing M2 polarization and enhancing inflammation-independent biological pathways. (**C**) Lactic acid acts as a signaling molecule. G protein-coupled receptor GPR81 is a lactic acid receptor found on both immune cells and cancer cells. Its activation in cancer cells promotes proliferation, drug resistance, and enhanced expression of PD-L1. Its activation on DCs is associated with decreased levels of cAMP, IL-6, IL-12, and suppressed antigen (Ag) presentation.

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
