# Peer review of "Lactic Acid and an Acidic Tumor Microenvironment suppress Anticancer Immunity"

_ijms, 2020, doi:10.3390/ijms21218363_

Round 1
Reviewer 1 Report
In this review, the authors summarized recent evidence that excess lactic acid in the TME was responsible for suppressing anticancer immunity. Novel therapeutic strategy targeting acidic TME would be a promising treatment as a combination therapy. The concept of current review is intriguing and the manuscript appears to be generally well-written. After minor revisions, this review would be acceptable for publishing in this journal.
Specific Comments
- The authors showed how each immune cells were affected by lactic acids. However, the description of the precise mechanism how acidic tumor microenvironment affect the characteristics of cancer cells would be insufficient. Additional explanation should be required in this manuscript.
- In addition, various enzymes and transporters are important for lactic acid production from cancer cells. These evidence should be summarized in a Figure for better understanding.
- In Figure 2B and C, the description of intracellular signaling and mechanism is not enough. More precise explanation would be needed for readers.
Author Response
In this review, the authors summarized recent evidence that excess lactic acid in the TME was responsible for suppressing anticancer immunity. Novel therapeutic strategy targeting acidic TME would be a promising treatment as a combination therapy. The concept of current review is intriguing and the manuscript appears to be generally well-written. After minor revisions, this review would be acceptable for publishing in this journal.
We thank the reviewer for the well-considered comments, and appreciate that the reviewer finds the concept of this current review intriguing
Point 1:The authors showed how each immune cells were affected by lactic acids. However, the description of the precise mechanism how acidic tumor microenvironment affect the characteristics of cancer cells would be insufficient. Additional explanation should be required in this manuscript.
Response 1: We agree with the reviewer’s comment. However, the primary focus of this review is to summarize how cancer-associated immune cells are affected by lactic acid. The effect of lactic acid on cancer cells is not our current emphasis, and there are a number of great reviews elsewhere on the topic. After discussing with the co-authors, we all feel that we should stay within our current focus, which is more exciting, more important, and less studied.
Point 2: In addition, various enzymes and transporters are important for lactic acid production from cancer cells. These evidences should be summarized in a Figure for better understanding.
Response 2: This would be a wonderful suggestion if we were to focus on lactic acid metabolism and the pathways involved (e.g. glycolysis, glutaminolysis, and the various enzymes/regulators). However, as we mentioned above, our focus is on how cancer-associated immune cells are affected by lactic acid as opposed to the mechanisms involved in lactic acid generation.
Point 3: In Figure 2B and C, the description of intracellular signaling and mechanism is not enough. More precise explanation would be needed for readers.
Response 3: We agree with the reviewer and have expanded the legend to Figure 2B and C accordingly to explain the mechanisms in more detail. See lines 355-361.
Reviewer 2 Report
In this review, Wang et al. summarized current research advances on the role of lactic acid and perhaps acidic tumor microenvironment (TME) in anticancer immunity. They discussed the diverse effects of increased lactic acid in TME on different immune cells. Moreover, several molecular mechanisms through which lactic acid acts on immune cells were outlined. Overall, this review is well written and organized, providing a clear snapshot on the relationship between acidic tumor microenvironment (TME) and anticancer immunity. However, I have a few comments on this review as following:
- The authors focused on the dark side of lactic acid in anticancer immunity. I wonder whether lactic acid, as a physiological metabolite, has any opposite effect on anticancer immunity. At least, I noticed a recent paper (PMID: 33095157, DOI: 10.7554/eLife.59996) showing a benefit of increased lactate on anti-tumor immunity through CD8+ T cells.
- Although the function/phenotype part is well reviewed, the molecular mechanism part is relative weak. What determines the different response of immune cells to lactic acid? How is the transcriptional/epigenomic network orchestrated in the immune cells in acidic TME?
- In t several places, the authors mentioned that there were opportunities and challenges of utilizing the new knowledge in fighting cancer. However, the authors did not provide any detailed hypothesis or possibility about how to translate these knowledge in the immunotherapy and what the major challenges or gaps are in achieving these aims.
Author Response
In this review, Wang et al. summarized current research advances on the role of lactic acid and perhaps acidic tumor microenvironment (TME) in anticancer immunity. They discussed the diverse effects of increased lactic acid in TME on different immune cells. Moreover, several molecular mechanisms through which lactic acid acts on immune cells were outlined. Overall, this review is well written and organized, providing a clear snapshot on the relationship between acidic tumor microenvironment (TME) and anticancer immunity. However, I have a few comments on this review as following:
Thanks! We appreciate the comment that “Overall, this review is well written and organized, providing a clear snapshot on the relationship between acidic tumor microenvironment (TME) and anticancer immunity”.
Point 1: The authors focused on the dark side of lactic acid in anticancer immunity. I wonder whether lactic acid, as a physiological metabolite, has any opposite effect on anticancer immunity. At least, I noticed a recent paper (PMID: 33095157, DOI: 10.7554/eLife.59996) showing a benefit of increased lactate on anti-tumor immunity through CD8+ T cells.
Response 1: This is a great point. In our review, we chose to focus on how lactic acid affects anti-cancer immunity under a chronic pathological condition (i.e. an acidic TME as generated by altered cancer metabolism). Although it is possible that lactic acid, along with other metabolites, can interact with immune cells differently under acute conditions such as exercise, it is outside of the focus of this review.
Point 2: Although the function/phenotype part is well reviewed, the molecular mechanism part is relative weak. What determines the different response of immune cells to lactic acid? How is the transcriptional/epigenomic network orchestrated in the immune cells in acidic TME?
Response 2: This is a great comment and it is fair. It is well documented that lactic acid and the acidic TME significantly impacts anti-cancer immunity. However, there are very few studies on the specific mechanisms of action beyond broad suggestions of which pathways may be involved (e.g. hypoxia, arginine metabolism). Indeed, this is a largely forgotten area of research. One of our major objectives for this review is to stimulate greater interest for more in-depth studies.
Point 3: In several places, the authors mentioned that there were opportunities and challenges of utilizing the new knowledge in fighting cancer. However, the authors did not provide any detailed hypothesis or possibility about how to translate these knowledge in the immunotherapy and what the major challenges or gaps are in achieving these aims.
Response 3: We appreciate the reviewer’s comments. Accordingly, we outlined in greater details some of the more common approaches that attempt to therapeutically reverse the acidic TME, as well as the major hurdles involved. See lines 316-324.